# Enterprise Implementation of Educational Technology: Exploring Employee Learning Behavior in E-Learning Environments

**Ching-Yeh Tsai [1,*]** and **Der-Chiang Li [2]**

1 Institute of Information Management, National Cheng Kung University, Tainan 70101, Taiwan
2 Chung Kung Research Center, Department of Industrial and Information Management, National Cheng Kung University, Tainan 70101, Taiwan; lidc@mail.ncku.edu.tw
* Correspondence: r78001025@gs.ncku.edu.tw or r78114014@gs.ncku.edu.tw

**Abstract:** In the rapidly evolving landscape of information technology and with the ubiquitousness of the internet, corporations are increasingly focused on harnessing educational technology to boost their competitive prowess. A pivotal question emerges: Can they implement this technology effectively and sustainably to enhance the learning capabilities of their workforce and facilitate their accumulation of knowledge and skills? This concern remains a central focus in the corporate sphere. From educational psychology, goal orientation theory offers an explanatory framework for understanding learner (employee) behavior variations following learning interventions. This research is grounded in the e-learning environment fostered by educational technology within corporations. It explores and hypothesizes the impact of employee achievement motivations—including learning, proving, and avoiding goal orientations—on self-regulated learning (encompassing cognitive, motivational, and behavioral adjustments). Additionally, this study examines how employees' self-regulated learning and learning satisfaction with the learning process influence learning effectiveness (learning outcomes) assessments. Our empirical survey targeted 380 employees from 26 companies participating in corporate educational technology learning (e-learning), with our research hypotheses tested through PLS structural equation modeling. The analysis indicates that employees' learning and proving goal orientations indirectly positively affect their learning outcomes by mediating self-regulated learning and learning satisfaction. Conversely, employees' avoidance goal orientation indirectly negatively impacts their learning outcomes by mediating their self-regulated learning and learning satisfaction. Finally, the researchers offer recommendations for management and future research directions.

**Keywords:** sustainability; e-learning; goal orientation; self-regulated learning; learning satisfaction; learning effectiveness evaluation; learning outcomes

## 1. Introduction

In today's rapidly changing business environment, e-learning within educational technology has become an indispensable core strategy for modern enterprises. With continuous advancements in information technology and deepening globalization, e-learning transcends traditional teaching and training methods, becoming a necessary condition for enterprises to maintain their competitiveness in the face of swift market changes [1–3]. The flexible application of technology education offers employees a learning mode that overcomes the limitations of time and location while also bringing significant benefits to enterprises by controlling their learning costs, enhancing employee performance, and achieving cross-national and remote training. By integrating educational technology, businesses have been able to smoothly transition from traditional teaching models to the new era of digital learning, fully leveraging the multiple advantages of e-learning, such as integrating synchronous and asynchronous learning methods and information technology, effectively promoting internal knowledge-sharing and exchange [3–5]. This enhances the support function and effectiveness of employee learning and meets the diverse needs of employee training, career development, and corporate growth.

However, differences still exist between employee learning behaviors and learning outcomes in e-learning environments, which remain a focus of attention for educators, training and human resources (HR) experts, and scholars. Originating from educational psychology, goal orientation theory provides a practical framework for explaining individual motivations, behaviors, and outcomes in learning [6]. This theory has been empirically supported in the fields of educational learning and business practice management [7–13], and its framework has been proven applicable to e-learning environments [14,15]. Individuals with different goal orientations pursue different objectives, such as long-term goals for self-improvement, short-term goals for demonstrating abilities, or avoidance responses in the face of setbacks, all of which affect their cognitive, emotional, and behavioral responses [8,16].

Therefore, in the corporate e-learning environment, guiding employees to set appropriate learning goals is crucial in enhancing the learning process and its outcomes. Beyond goal orientation theory, the self-regulated learning theory has also attracted widespread attention from the academic community and researchers, and it is considered a critical factor in enhancing individual learning motivation and improving learning effectiveness in both traditional and e-learning environments [17–21]. Self-regulated learning not only involves the learners' cognition, emotions, and behaviors but also delves into how these factors affect their selection and application of self-regulated learning strategies and their close connection to post-learning outcomes [17,20–23]. Since the learners themselves control self-regulated learning and directly impact their post-learning performance behaviors, many studies have explored the relationship between these factors and learning outcomes [23–28].

This study explores how employees' goal orientations in an e-learning environment affect their self-regulated learning behaviors, and how these self-regulated learning strategies further affect overall learning effectiveness (learning satisfaction and learning outcomes). Additionally, based on Keller's [29] ARCS motivation model, learning motivation must simultaneously consider four elements: attention, relevance, confidence, and satisfaction. These elements are deemed crucial for stimulating learners' motivation and are closely related to learning satisfaction and outcomes [30,31]. Therefore, this study will also explore how employees' learning satisfaction in an e-learning environment affects their learning outcomes.

In summary, the main objectives of this study are as follows:

To explore how employees' goal orientations in an e-learning environment affect their self-regulated learning strategies.

To verify how employees' self-regulated learning strategies in an e-learning environment affect their learning satisfaction and outcomes.

To examine the impact of employees' learning satisfaction on learning outcomes in an e-learning environment.

To infer and verify whether employees' goal orientations indirectly affect their learning outcomes through the mediating roles of self-regulated learning and learning satisfaction.

This study provides practical suggestions for enterprises to implement e-learning strategies more effectively, enhancing employee satisfaction and their learning outcomes.

## 2. Literature Review

### 2.1. Goal Orientation in E-Learning Environments

Goal orientation theory, rooted in educational psychology, was initially used to explain learners' motivations and behaviors in traditional educational settings [7,10]. This theory emphasizes how an individual's achievement motivation can influence their behavioral orientation, subsequently affecting their learning outcomes [6,14,22]. Over time, the influence of goal orientation theory has extended beyond the educational realm into organizational management and industrial psychology [10,32–35]. In e-learning environments, goal orientation theory is beneficial for understanding how learners regulate their learning process by setting and pursuing different types of learning goals [14,36]. These goal types include a learning goal orientation, proving goal orientation, and avoiding goal orientation [10,22,37].

Learners with a learning goal orientation tend to seek a mastery of knowledge and skills, focusing more on self-improvement and the learning process itself rather than mere outcomes [8,38]. These learners believe that abilities can be enhanced through effort and learning, valuing challenges and effort [8]. Therefore, in e-learning environments, this goal orientation helps learners adapt to new technologies and methods, performing better in the learning process [12–14,39].

Unlike the learning goal orientation, learners with a proving goal orientation focus more on demonstrating their capabilities in front of others. Their learning motivation is related to external rewards or recognition, and they feel more significant pressure in highly competitive environments [12,39–41]. Such learners in e-learning environments might lean towards surface learning strategies, such as repetition and memorization, as they seek short-term grades and others' acknowledgment [40–42].

Learners with an avoiding goal orientation focus more on avoiding failure and negative evaluation [23,26]. These learners typically exhibit an intense fear of failure in learning environments and may retreat when faced with challenges [26,43,44]. In e-learning contexts, this may lead them to avoid difficult tasks and seek the smallest effort possible to complete learning tasks [22,26,44].

In e-learning environments, learners' goal orientations not only affect their learning behaviors but also their adaptation to technology and overall participation in the learning process. Learners with a learning goal orientation tend to choose deep learning strategies, such as concept integration and critical thinking. In contrast, those with proving and avoiding goal orientations might lean towards surface learning or avoiding challenging tasks [14].

Implementing e-learning in corporate organizations requires understanding the different types of goal orientations to design effective corporate training programs. This helps enhance employees' learning motivation and engagement, ensuring that training programs align with employees' personal and professional goals. For instance, employees with a learning goal orientation might be more suited to training that is focused on skill enhancement and innovative thinking. In contrast, those with a proving goal orientation might be more suited to training that allows them to demonstrate their capabilities and gain recognition [14,15,32].

Overall, applying goal orientation theory in e-learning environments helps us understand how learners' goal settings influence their learning behaviors and outcomes and guide them in choosing appropriate learning strategies. These theories are significant for designing effective e-learning courses and training programs, especially in fostering employees' learning motivation and engagement [14,15]. Thus, understanding and applying goal orientation theory is crucial for improving the learning process in e-learning environments.

### 2.2. Self-Regulated Learning in Employee Education

In today's rapidly changing workplace environment, self-regulated learning (SRL) plays a crucial role in employee education. Particularly in the context of e-learning, the significance of self-regulated learning is increasingly evident. This learning approach involves how learners actively participate in and control their learning process, which is critical in online learning scenarios that lack direct guidance and support from traditional face-to-face environments [23–28].

Educational psychologists initially proposed self-regulated learning to describe how learners use strategies to control their learning process [44–46]. This process includes cognitive, motivational, and behavioral aspects. Therefore, learners can set goals, select appropriate learning strategies, and monitor and adjust their learning journey through self-reflection [22,28].

As technology and work environments constantly evolve in corporate education and training, employees must quickly adapt to new knowledge and methods. Self-regulated learning allows employees to manage their learning progress more effectively, maintaining

competitiveness in a dynamic work environment. This learning approach significantly enhances employees' professional skills and promotes their personal development [28].

Self-regulated learning is divided into three core components: cognitive regulation, motivational regulation, and behavioral regulation [22,24,42].

Cognitive regulation involves how learners process and remember learning materials. Effective cognitive regulation strategies, such as rehearsal, elaboration, organization, and monitoring comprehension, help employees process information more efficiently in e-learning environments [22,46–48].

Motivational regulation involves learners' intrinsic motivation and extrinsic incentives. Strategies like setting challenging goals, self-encouragement, and appropriate attribution help increase learners' engagement and motivation [22,26,47,48].

Behavioral regulation is related to learners' actual actions, including effort and persistence. In e-learning environments, behavioral regulation is particularly important as learners need to self-manage to adapt to online learning requirements [22,42,47,48].

The impact of self-regulated learning on learning outcomes has been validated in numerous studies [44,49–52]. Learners who effectively use self-regulated learning strategies often achieve higher learning efficiency and a deeper understanding of knowledge. These learners perform better in handling complex tasks and challenges and gain higher satisfaction from the learning process [44,50]. In a corporate environment, this means employees with strong self-regulated learning abilities are more likely to quickly master new skills and improve their work efficiency, positively impacting the organization's overall performance [53].

In summary, the importance of self-regulated learning in employee education and training is undeniable. To facilitate employees' professional development, corporations and educational institutions should design training programs and materials that support self-regulated learning. This includes providing strategy guidance, self-assessment tools, and ongoing feedback mechanisms. Additionally, encouraging employees to set specific and achievable learning goals and assess their learning progress through self-monitoring will help them improve their learning outcomes and career achievements. Through such efforts, corporations can promote employees' personal growth and enhance the organization's competitiveness and adaptability.

### 2.3. The Relationships between Goal Orientation, Self-Regulated Learning, Learning Satisfaction, and Learning Outcomes

In modern educational and corporate training environments, especially within the context of e-learning, understanding the impact of employees' goal orientations and self-regulated learning on overall learning effectiveness is crucial. This study differentiates overall learning effectiveness into short-term learning satisfaction and mid-to-long-term learning outcomes [54], which are explained as follows.

Learning satisfaction refers to learners' overall feelings or attitudes towards their learning process and outcomes [54,55]. According to Baker and Crompton [55], learning satisfaction involves the learners' psychological responses to the learning process and outcomes, including positive emotions and satisfaction levels. Learning satisfaction is considered a key indicator for measuring the quality of the learning experience in both educational and corporate training fields [28,56]. It is crucial in e-learning environments, as it affects learners' continued engagement with and effective utilization of learning resources and serves as an indicator for assessing learning performance [28,56,57].

Learning outcomes refer to the extent to which learners can effectively and sustainably apply the knowledge and skills acquired through learning in their actual work [54,58,59]. It serves as an important indicator for assessing learning performance, primarily aimed at understanding individuals' learning states, i.e., the final results after learning [54,59]. It also serves as a crucial basis for educators to improve teaching, training, and the learners themselves to enhance learning [59]. Learning is a process facilitated by activities and experiences that evolve behavior; learning outcomes are measured by the degree of exter-

nal behavioral or performance changes after participation in learning activities, and are considered a critical indicator of learning [59,60].

The following will delve into the interactions between employees' goal orientations, self-regulated learning, learning satisfaction, and learning outcomes, and their impact on overall learning effectiveness.

### 2.3.1. The Relationship between Goal Orientation, Self-Regulated Learning, and Learning Satisfaction

Goal orientation theory and self-regulated learning provide significant perspectives from which to understand learning satisfaction. Through goal orientation theory and self-regulated learning strategies, one can further comprehend how individuals set and pursue learning goals and utilize relevant learning strategies to achieve outcomes and gain satisfaction from the learning process [7,10,22–28,37]. Learners with a learning goal orientation tend to pursue a deep understanding and mastery of skills; this intrinsic motivation promotes their satisfaction with the learning process [8,38]. Simultaneously, they effectively use self-regulated learning strategies, such as cognitive, motivational, and behavioral adjustments, enabling these learners to master learning materials more effectively, thereby increasing their satisfaction with the learning process and making them more inclined to invest more effort and time to ensure effective learning [22,26,47,51].

Learners with a proving goal orientation are more concerned with demonstrating their abilities to others [12,39]. Therefore, they are more inclined to adopt superficial self-regulated learning strategies, relying on surface learning driven by external rewards and punishments for cognitive and motivational adjustments [7,22,26,42,45]; behaviorally, they are proactive in ensuring the best results within a limited time [22,47,48]. Conversely, learners with an avoiding goal orientation primarily aim to avoid poor performance or criticism [22,41]. Thus, their use of self-regulated learning strategies differs from the other two types. In terms of cognitive and motivational adjustments, they adopt a more passive learning approach and often do not know how to respond when facing difficulties and challenges [23,44]. Behaviorally, they tend to avoid difficult tasks and seek the smallest effort possible to complete learning tasks [22,44].

Self-regulated learning theory is commonly used to explain how learners set goals, plan, self-monitor, and adjust the learning process to achieve the best learning outcomes [44–46,51]. Past findings have shown a positive correlation between self-regulated learning and learners' motivation, engagement, and learning outcomes [28,50,61]. When learners effectively use self-regulated learning strategies, they often achieve reasonable learning satisfaction and are better able to master acquired knowledge and skills [28,50,53].

Furthermore, when learners effectively use cognitive adjustments, they feel a sense of achievement, thereby increasing their learning satisfaction [53,61,62]. In terms of motivational adjustments, when learners have strong intrinsic motivation and external incentives, they are more fully engaged in learning, thus improving their learning satisfaction [61,62]. Finally, in behavioral adjustments, learners effectively manage learning resources through actual action to achieve their learning goals, gaining high learning satisfaction [53,61].

In summary, this study infers that in the e-learning environment, employees' goal orientations (learning orientation, proving orientation, avoiding orientation) are related to self-regulated learning (cognitive adjustment, motivational adjustment, behavioral adjustment). Employees' self-regulated learning (cognitive adjustment, motivational adjustment, behavioral adjustment) is closely related to their learning satisfaction.

### 2.3.2. The Impact of Self-Regulated Learning and Learning Satisfaction on Learning Outcomes

What factors influence learners' learning outcomes? Some scholars believe self-regulated learning is the best predictive variable of learning outcomes [44,50,63]. Research also suggests that learning satisfaction is considered one of the crucial factors affecting learning outcomes [30,64,65]. Most of these studies focus on students, using academic grades as learning outcomes, to explore the impact of different self-regulated learning

strategies. Studies have shown that strategies such as cognitive, motivational, and behavioral adjustments significantly positively affect learning outcomes [44,49–52]. Furthermore, research involving corporate employees also indicates the significant positive effects of self-regulated learning on learning outcomes [63,66]. In other words, learners proficient in using self-regulated learning strategies tend to achieve better learning outcomes.

Moreover, theoretical analysis and past research indicate that learning satisfaction not only affects students' learning motivation and engagement but also their learning effectiveness [31,56,64,65]. According to Kim and Park [30], learning satisfaction in an e-learning environment significantly positively impacts learning outcomes. When learners are satisfied with their learning experience, they are more likely to engage in learning activities and focus more on their learning, ultimately achieving better learning outcomes [30,50,56,57,67]. Conversely, when learners are dissatisfied with their learning experience, their motivation may decrease, leading to less effective learning outcomes.

In summary, in an e-learning environment, besides learners' self-regulated learning affecting learning outcomes, learners' learning satisfaction also positively correlates with their learning outcomes.

### 2.3.3. The Interaction between Goal Orientation, Self-Regulated Learning, and Overall Learning Effectiveness

Research by Artino [67] mentions that learners who extensively adopt self-regulated learning have higher self-efficacy than those with lesser capabilities in using such strategies. They believe they can utilize self-efficacy to aid their learning and employ self-regulated learning to enhance their learning satisfaction and outcomes [28,64,67]. Additionally, some studies exploring the relationship between learning goals, motivation, behavior, and outcomes have found self-regulated learning to play a crucial mediating role [52,66,68,69], exerting a significant impact on learning outcomes [22,52,70].

Furthermore, Snyder, Raben, and Farr's [71] study suggests that learning satisfaction can serve as a mediator between learning behaviors and outcomes [28,56]. When learners effectively use self-regulated learning strategies and persistently and continuously invest effort in learning, not only can they enhance their learning motivation but they also gain satisfaction from the learning process, thereby improving their learning outcomes [65].

In conclusion, this study posits that, in the e-learning process, employees' goal setting (learning goals) provides direction for their learning behaviors (self-regulated learning), and the choice of these behaviors and the extent of the personal effort enhance short-term learning effectiveness (learning satisfaction), ultimately achieving mid-to-long-term learning effectiveness (learning outcomes). Therefore, this study infers that, in the e-learning environment, employees' goal orientation indirectly affects their overall learning effectiveness (including learning satisfaction and outcomes) through self-regulated learning. Furthermore, employees' self-regulated learning indirectly affects their learning outcomes through learning satisfaction. Hence, employees' self-regulated learning and learning satisfaction play significant mediating roles in the impact of goal orientation on learning outcomes.

### 3. Hypotheses and Research Methodology

#### 3.1. Hypotheses

Based on the literature review of the relationships between goal orientation, self-regulated learning, and learning satisfaction, this study proposes the following hypotheses:

H1: Employees' learning goal orientation significantly positively influences their (a) cognitive regulation, (b) motivational regulation, and (c) behavioral regulation.

H2: Employees' proving goal orientation significantly positively influences their (a) cognitive regulation, (b) motivational regulation, and (c) behavioral regulation.

H3: Employees' avoiding goal orientation significantly negatively influences their (a) cognitive regulation, (b) motivational regulation, and (c) behavioral regulation.

H4: Employees' (a) cognitive regulation, (b) motivational regulation, and (c) behavioral regulation have a significant positive effect on their learning satisfaction.

Through the analysis of how self-regulated learning and learning satisfaction impact learning outcomes, it is indicated that learners' self-regulated learning and learning satisfaction influence their learning outcomes. Thus, the study proposes the following hypotheses:

H5: Employees' (a) cognitive regulation, (b) motivational regulation, and (c) behavioral regulation significantly enhance their learning outcomes.

H6: Employees' learning satisfaction has a significant positive impact on their learning outcomes.

Integrating an analysis of the interactions between goal orientation, self-regulated learning, and overall learning effectiveness, this study infers that employees' goal orientations indirectly affect their learning satisfaction and learning outcomes through the mediating effects of self-regulated learning strategies. It proposes the following hypotheses:

H7: Employees' goal orientations indirectly influence their learning satisfaction and learning outcomes through the mediating effects of self-regulated learning strategies.

H8: Employees' self-regulated learning strategies indirectly influence their learning outcomes through the mediating effects of learning satisfaction.

*3.2. Research Framework*

Drawing upon our research motivation and hypotheses, this study proposes the conceptual research framework shown in Figure 1.

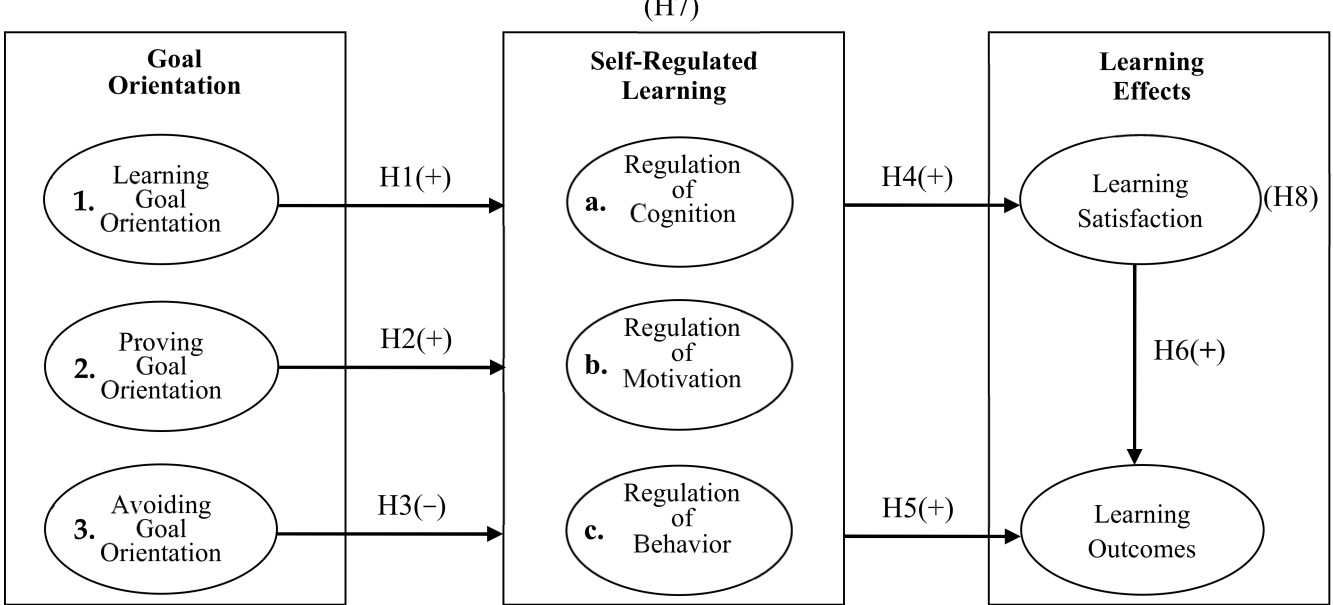

**Figure 1.** Conceptual research framework.

*3.3. Research Participants and Data Collection*

This study's hypotheses were primarily tested through a survey targeting employees of small and medium-sized enterprises (SMEs) in Taiwan that have implemented e-learning platforms, explicitly focusing on employees with more than six months of experience using the "e-learning platform." To ensure that employees had established a stable pattern of using (or not using) the e-learning system, following the recommendation of Eggert and Serdaroglu [72], employees with at least six months of system use were selected for this formal analysis. The survey was conducted using convenience sampling and collected via physical questionnaires and Google Forms. The survey was initiated on 1 July 2023, and concluded on 31 July 2023, with data collected anonymously. A total of 800 questionnaires were distributed across 40 companies, averaging about 20 per company (with fewer than 200 employees per company), resulting in 425 returned questionnaires from 26 companies (a response rate of 53.13%). After excluding incomplete responses and those with less than

six months of experience, 380 valid questionnaires were obtained, yielding an effective response rate of 47.5%.

Most respondents came from the manufacturing industry, which accounted for 53.85% of responses. Regarding the department affiliation of the 380 employees, the administrative, manufacturing, business, and shipping departments represented 10.00%, 18.68%, 14.47%, and 12.63%, respectively, with a total response rate across these departments of 55.78%. The sample structure of the companies and the departments of the employees are detailed in Table 1.

**Table 1.** Structure of the companies (N = 26) and number of employees in the departments (N = 380).

| Statistical Variables of Industries and Departments | | Number (%) |
|---|---|---|
| Classification of industries | Manufacturing industry | 14(53.85%) |
| | Trading industry | 8(30.77%) |
| | Circulation industry | 4(15.38%) |
| Type of department | Administrative dept. | 38 (10.00%) |
| | Manufacturing dept. | 71 (18.68%) |
| | Business dept. | 55 (14.47%) |
| | Human resources dept. | 29 (07.63%) |
| | R&D dept. | 26 (06.84%) |
| | Finance dept. | 31 (08.16%) |
| | Procurement dept. | 37 (09.74%) |
| | Management dept. | 27 (07.11%) |
| | Shipping dept. | 48 (12.63%) |
| | General dept. | 18 (04.74%) |

In the sample structure, females were predominant, comprising 53.68% of the sample. The primary users of e-learning were employees aged 36 and below (88.95%), with the majority holding a university degree or higher (70.00%). The majority were unmarried (62.11%), with e-learning experience mainly within the last two years (87.63%), and over three years work experience (68.42%). Demographic details of the sample are presented in Table 2.

**Table 2.** Sample structure (N = 380).

| Demographic Variables | | Number (%) |
|---|---|---|
| Gender | Male | 176 (46.32%) |
| | Female | 204 (53.68%) |
| Age | 25 years old and below | 123(32.37%) |
| | 26~30 years old | 102 (26.84%) |
| | 31~35 years old | 113 (29.74%) |
| | Over 36 years old | 42 (11.05%) |
| Education level | Senior high school and below | 32(08.42%) |
| | Junior college | 56 (14.74%) |
| | College degree | 266 (70.00%) |
| | Master's degree | 26 (06.84%) |
| Marital status | Unmarried | 236 (62.11%) |
| | Married | 144 (37.89%) |
| Experience with e-learning | One year and below | 197(51.84%) |
| | 1~2 years | 136 (35.78%) |
| | 2~4 years | 41 (10.79%) |
| | More than 4 years | 6 (01.58%) |
| Work experience | Three years and below | 76 (20.00%) |
| | 3~5 years | 137 (36.05%) |
| | 5~10 years | 123 (32.37%) |
| | More than 10 years | 44 (11.58%) |

Given the convenience sampling method, there is a potential risk of sampling bias. However, the diversity in the sample's sources (departments, gender, age groups, work experience, etc.), as shown in Tables 1 and 2, suggests that the sample is diverse. Further statistical analyses (T-tests and ANOVA) showed no significant differences in perceptions of the study variables among participants from different areas. Lastly, early respondents (those from the first week) and late respondents (those from the fourth week) were compared to detect any differences in their perceptions of the study variables, with the analysis showing no significant differences. Overall, the returned questionnaires are considered to have appropriate representativeness.

*3.4. Questionnaire Design*

The questionnaire comprises five sections, including participants' goal orientation, self-regulated learning, learning satisfaction, learning outcomes, and demographic variables (as shown in Table 3). Each scale utilizes a Likert seven-point scale, where "1" indicates strong disagreement and "7" represents strong agreement. The sections are described below:

**Table 3.** Construct and References.

| Construct | Research Variable | Items | References |
|---|---|---|---|
| Goal orientation | Learning goal orientation | 7 | VandeWalle [37] Elliot and Church [70] Pintrich [22] |
| | Proving goal orientation | 6 | |
| | Avoiding goal orientation | 5 | |
| Self-regulated learning | Cognitive regulation | 6 | Pintrich [22] Gordon et al. [47] Bouffard et al. [48] |
| | Motivational regulation | 6 | |
| | Behavioral regulation | 6 | |
| Learning satisfaction | | 6 | Kuo et al. [53] |
| Learning outcomes | | 5 | Alavi et al [58] Pike et al. [60] |
| Demographic variables | | | |

Goal orientation scale: This study utilized the goal orientation scale designed by VandeWalle [37], Elliot and Church [70], and Pintrich [22], allowing employees to self-assess their learning goal orientation (seven items), proving goal orientation (six items), and avoiding goal orientation (five items). Higher scores indicate a stronger inclination toward that specific goal orientation. An example of a questionnaire item is "Spending considerable time learning new skills for my job is worthwhile."

Self-regulated learning scale: The self-regulated learning scale, adapted from Pintrich [22], Gordon, Lindner, and Harris [47], and Bouffard, Boisvert, Vezeau, and Larouche [48], was employed. This scale includes cognitive regulation (six items), motivational regulation (six items), and behavioral regulation (six items); higher scores denote a greater tendency toward that learning strategy. An example of a questionnaire item is "I set stage-specific goals for each phase of the learning activity."

Learning satisfaction scale: The study used the learner satisfaction scale developed by Kuo, Walker, Schroder, and Belland [53] (six items) to measure employee satisfaction during the learning process. Higher scores indicate a greater perceived level of learner satisfaction. An example of a questionnaire item was "Overall, I am satisfied with the content of the e-learning course."

Learning outcomes scale: This study utilized the learning outcomes scale created by Pike, Kuh, McCormick, Ethington, and Smart [60] and Alavi, Marakas, and Yoo [58] (five items) to measure the personal achievements of employees post e-learning. Higher scores suggest better-perceived learning outcomes. An example of a questionnaire item

was "After undergoing the e-learning course, my ability to apply job skills and knowledge has been significantly enhanced."

Demographic variables include gender, age, education level, marital status, experience of e-learning, work experience, classification of industries, and type of department.

## 4. Results

### 4.1. Common Method Variance (CMV) Test

The use of self-report scales for collecting cognitive information from a single respondent can potentially lead to common method variance (CMV) bias, thereby possibly overestimating or underestimating the relationships between variables [6]. To mitigate the issue of CMV, this study incorporated the recommendations from Podsakoff, Mackenzie, Lee and Podsakoff [73] into its questionnaire design, including anonymous surveying and random item arrangement. Additionally, Harman's single-factor test was used to assess the presence of significant CMV [74]. The test revealed that the variance explained by the first factor was 36.85%, which is below the 50% threshold, indicating that severe CMV is unlikely in this study.

### 4.2. Verification of the Research Hypotheses

The research framework proposed in this article was analyzed using the partial least squares (PLS) method. PLS can simultaneously handle multiple research facets and variables and provides robust parameter estimation results in small samples, without requiring a multivariate normal distribution of the original data [75,76]. SPSS 20.0 and SmartPLS4 were used for the analysis.

#### 4.2.1. Outer Model

The outer model was used to examine the reliability, convergent validity, and discriminant validity of this study. The standardized factor loadings of each construct should exceed 0.5, otherwise it is deemed non-representative and is excluded [77]. Excluding four items (ROM5, ROM6, ROB6, and LO4) that did not meet the standard, the factor loadings of the remaining items exceeded 0.5 (Table 4); thus, they were retained.

**Table 4.** PLS scale analysis results.

| Construct | Research Variable | Factor Loading | Cronbach's $\alpha$ | Composite Reliability (CR) | AVE |
|---|---|---|---|---|---|
| Learning Goal Orientation (LGO) | LGO1 | 0.859 | 0.928 | 0.931 | 0.698 |
| | LGO2 | 0.815 | | | |
| | LGO3 | 0.870 | | | |
| | LGO4 | 0.877 | | | |
| | LGO5 | 0.824 | | | |
| | LGO6 | 0.830 | | | |
| | LGO7 | 0.769 | | | |
| Proving Goal Orientation (PGO) | PGO1 | 0.770 | 0.889 | 0.893 | 0.643 |
| | PGO2 | 0.860 | | | |
| | PGO3 | 0.769 | | | |
| | PGO4 | 0.788 | | | |
| | PGO5 | 0.801 | | | |
| | PGO6 | 0.819 | | | |

**Table 4.** *Cont.*

| Construct | Research Variable | Factor Loading | Cronbach's $\alpha$ | Composite Reliability (CR) | AVE |
|---|---|---|---|---|---|
| Avoiding Goal Orientation (AGO) | AGO1 | 0.767 | 0.857 | 0.920 | 0.637 |
| | AGO2 | 0.874 | | | |
| | AGO3 | 0.745 | | | |
| | AGO4 | 0.664 | | | |
| | AGO5 | 0.914 | | | |
| Regulation of Cognition (ROC) | ROC1 | 0.776 | 0.915 | 0.919 | 0.703 |
| | ROC2 | 0.822 | | | |
| | ROC3 | 0.885 | | | |
| | ROC4 | 0.882 | | | |
| | ROC5 | 0.869 | | | |
| | ROC6 | 0.792 | | | |
| Regulation of Motivation (ROM) | ROM1 | 0.834 | 0.881 | 0.883 | 0.737 |
| | ROM2 | 0.903 | | | |
| | ROM3 | 0.864 | | | |
| | ROM4 | 0.830 | | | |
| Regulation of Behavior (ROB) | ROB1 | 0.796 | 0.895 | 0.899 | 0.706 |
| | ROB2 | 0.876 | | | |
| | ROB3 | 0.794 | | | |
| | ROB4 | 0.845 | | | |
| | ROB5 | 0.887 | | | |
| Learning Satisfaction (LS) | LS1 | 0.849 | 0.930 | 0.930 | 0.742 |
| | LS2 | 0.908 | | | |
| | LS3 | 0.894 | | | |
| | LS4 | 0.820 | | | |
| | LS5 | 0.862 | | | |
| | LS6 | 0.832 | | | |
| Learning Outcomes (LO) | LO1 | 0.872 | 0.922 | 0.922 | 0.810 |
| | LO2 | 0.912 | | | |
| | LO3 | 0.911 | | | |
| | LO5 | 0.905 | | | |

In terms of reliability, the study applied Cronbach's alpha and composite reliability (CR) to measure internal consistency across constructs. A Cronbach's alpha value of at least 0.7 indicates credibility [78]. A higher CR value suggests a high correlation among items within a construct and higher internal consistency. Hair et al. [77] and Fornell and Larcker [79] recommended a CR value of above 0.6. The Cronbach's alpha and CR values of all constructs in this study were above the recommended levels, indicating their high internal consistency (Table 4).

Convergent validity refers to the degree to which multiple indicators of the same construct are convergent or related. According to Hair et al. [77] and Fornell and Larcker [79], convergent validity should meet the following criteria: (1) factor loadings of each construct of above 0.7; (2) CR of above 0.6; and (3) average variance extracted (AVE) of above 0.5. As shown in Table 4, this study exhibits convergent validity.

Discriminant validity examines the degree of differentiation between constructs in the outer model. The greater the differentiation between constructs, the lower their correlation, indicating discriminant validity. If the square root of a construct's AVE value is greater than its correlation coefficients with other constructs, it indicates adequate discriminant validity [79] (Table 5). Moreover, this study employed the heterotrait–monotrait (HTMT) ratio for assessing discriminant validity. HTMT values below the standard threshold of 0.9 indicate adequate discriminant validity [80]. As Table 5 shows, the square root of the AVE for any two constructs is greater than their correlation coefficient. Table 6 shows that the HTMT values between constructs in this study are below 0.9, demonstrating their sufficient discriminant validity.

**Table 5.** Matrix of means, standard deviations, and correlation coefficients of the latent constructs and AVE square roots.

| Factors | Mean | S.D. | LGO | PGO | AGO | ROC | ROM | ROB | LS | LO |
|---|---|---|---|---|---|---|---|---|---|---|
| **LGO** | 5.525 | 0.797 | **0.836** | | | | | | | |
| **PGO** | 4.908 | 0.872 | 0.549 | **0.802** | | | | | | |
| **AGO** | 3.757 | 1.036 | −0.206 | 0.083 | **0.798** | | | | | |
| **ROC** | 5.361 | 0.838 | 0.492 | 0.346 | −0.230 | **0.839** | | | | |
| **ROM** | 5.084 | 0.889 | 0.447 | 0.370 | −0.234 | 0.700 | **0.858** | | | |
| **ROB** | 5.387 | 0.850 | 0.527 | 0.418 | −0.207 | 0.795 | 0.798 | **0.840** | | |
| **LS** | 5.190 | 0.889 | 0.333 | 0.346 | −0.105 | 0.665 | 0.633 | 0.683 | **0.861** | |
| **LO** | 5.453 | 0.859 | 0.359 | 0.358 | −0.127 | 0.656 | 0.601 | 0.689 | 0.734 | **0.900** |

Note 1: Values in the diagonal line are the AVE square roots of each latent construct; all other values are the coefficients of correlation between the constructs. Note 2: LGO = learning goal orientation; PGO = proving goal orientation; AGO = avoiding goal orientation; ROC = regulation of cognition; ROM = regulation of motivation; ROB = regulation of behavior; LS = learning satisfaction; LO = learning outcomes.

**Table 6.** Results of discriminant validity by HTMT ratio.

| Factors | LGO | PGO | AGO | ROC | ROM | ROB | LS | LO |
|---|---|---|---|---|---|---|---|---|
| **LGO** | | | | | | | | |
| **PGO** | 0.587 | | | | | | | |
| **AGO** | 0.220 | 0.162 | | | | | | |
| **ROC** | 0.523 | 0.373 | 0.236 | | | | | |
| **ROM** | 0.493 | 0.414 | 0.240 | 0.778 | | | | |
| **ROB** | 0.573 | 0.456 | 0.216 | 0.874 | 0.898 | | | |
| **LS** | 0.355 | 0.375 | 0.107 | 0.720 | 0.697 | 0.747 | | |
| **LO** | 0.382 | 0.390 | 0.132 | 0.713 | 0.667 | 0.758 | 0.791 | |

Note 1: LGO = learning goal orientation; PGO = proving goal orientation; AGO = avoiding goal orientation; ROC = regulation of cognition; ROM = regulation of motivation; ROB = regulation of behavior; LS = learning satisfaction; LO = learning outcomes.

Regarding the model fit, we initially employed the global fit index GoF to calculate the overall indicator of the measurement and structural models, providing the overall predictive utility of the model [81]. The GoF is the geometric mean of the average communality and the average $R^2$. Indicator values of 0.1, 0.25, and 0.36 represent weak, moderate, and strong fits, respectively. Here, the GoF value was 0.531, indicating a well-fitting model.

$$\text{GoF} = \sqrt{\overline{\text{communality}} * \overline{R^2}} = \sqrt{0.710 * 0.396} = 0.531$$

Furthermore, the standardized root mean square residual (SRMR) was used to assess the fit of the theoretical structural model (goodness of fit). The obtained SRMR (0.060) was below the acceptable threshold (0.08) [82]. This outcome suggests a good fit of the structural model.

### 4.2.2. Inner Model and Testing of Hypotheses

In PLS, the path structure between constructs is called the inner model. The hypothesis testing and path analysis results of the inner model are presented in Figure 2 and Table 7.

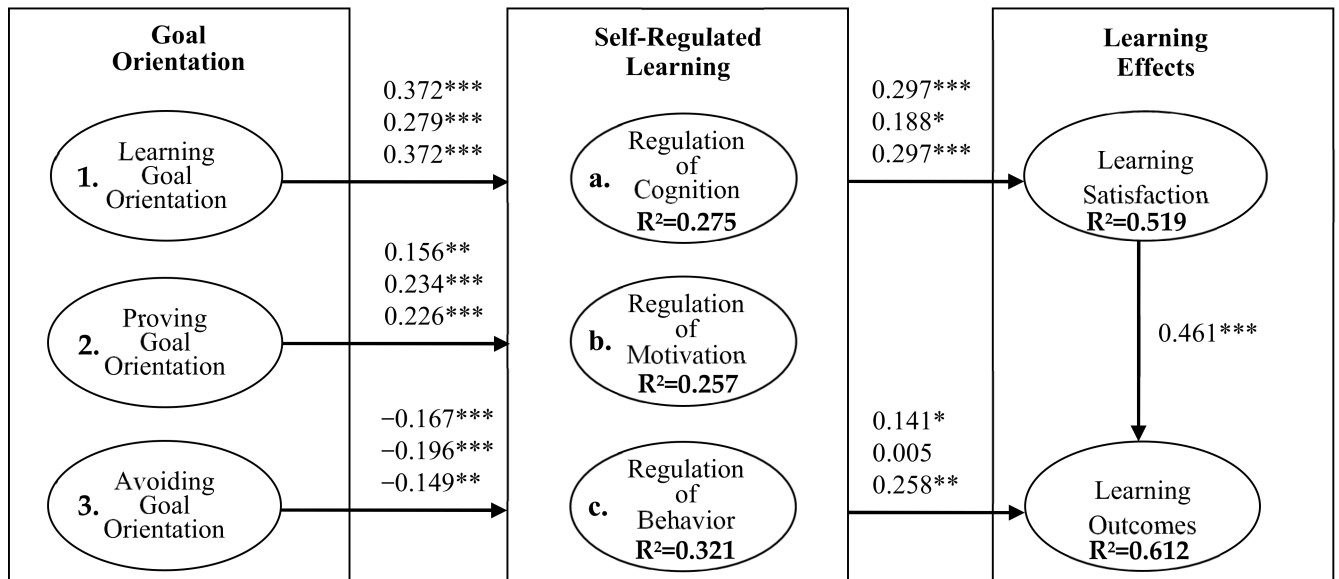

**Figure 2.** Standardized path coefficients and the significance of the inner model (*, *p*-value < 0.05; **, *p*-value < 0.01; ***, *p*-value < 0.001).

**Table 7.** Summary of the inner model results.

| | | Hypothesis | | | Path Coefficient | | t-Value | Result |
|---|---|---|---|---|---|---|---|---|
| | a(+): | LGO | → | ROC | 0.372 | *** | 6.026 | Supported |
| H1 | b(+): | LGO | → | ROM | 0.279 | *** | 4.209 | Supported |
| | c(+): | LGO | → | ROB | 0.372 | *** | 6.094 | Supported |
| | a(+): | PGO | → | ROC | 0.156 | ** | 2.968 | Supported |
| H2 | b(+): | PGO | → | ROM | 0.234 | *** | 4.242 | Supported |
| | c(+): | PGO | → | ROB | 0.226 | *** | 4.495 | Supported |
| | a(−): | AGO | → | ROC | −0.167 | *** | 3.191 | Supported |
| H3 | b(−): | AGO | → | ROM | −0.196 | *** | 3.435 | Supported |
| | c(−): | AGO | → | ROB | −0.149 | ** | 2.732 | Supported |
| | a(+): | ROC | → | LS | 0.297 | *** | 4.121 | Supported |
| H4 | b(+): | ROM | → | LS | 0.188 | * | 2.432 | Supported |
| | c(+): | ROB | → | LS | 0.297 | *** | 3.302 | Supported |
| | a(+): | ROC | → | LO | 0.141 | * | 2.250 | Supported |
| H5 | b(+): | ROM | → | LO | 0.005 | | 0.068 | Not Supported |
| | c(+): | ROB | → | LO | 0.258 | ** | 2.951 | Supported |
| H6(+): | | LS | → | LO | 0.461 | *** | 9.902 | Supported |

Note 1: LGO = learning goal orientation; PGO = proving goal orientation; AGO = avoiding goal orientation; ROC = regulation of cognition; ROM = regulation of motivation; ROB = regulation of behavior; LS = learning satisfaction; LO = learning outcomes. Note 2: *, *p*-value < 0.05; **, *p*-value < 0.01; ***, *p*-value < 0.001. Note 3: Number of bootstrap samples = 5000.

### 4.3. Verification of Hypotheses

The analysis results in Table 7 show that (1) employees' learning and proving goal orientations have a significant positive impact on their self-regulated learning, while their avoiding goal orientation significantly negatively impacts their self-regulated learning. Therefore, hypotheses 1abc, 2abc, and 3abc are supported. (2) Employees' three types of self-regulated learning have a significant positive impact on their learning satisfaction.

Thus, hypothesis 4abc are supported. (3) Employees' cognitive and behavioral regulation have a significant positive impact on their learning outcomes, while motivational regulation has a positive impact on their learning outcomes but is not significant. Therefore, hypothesis 5ac are supported, while hypothesis 5b is not supported. (4) Employees' learning satisfaction has a significant positive impact on their learning outcomes. Thus, hypothesis 6 is supported.

### 4.4. Mediation Effects Analysis

To evaluate the proposed mediation model, besides using the Sobel [83] Z-test, we also considered our empirical results and the recommendations from Tofighi and MacKinnon [84] and MacKinnon, Coxe, and Baraldi [85]. The product of the distribution method was used to calculate the mediating effects of self-regulated learning strategies and learning satisfaction, along with their confidence intervals (https://amplab.shinyapps.io/MEDCI) (accessed on 30 October 2023).

We examined the influence of employees' goal orientations on both their learning satisfaction and outcomes by considering two mediators: self-regulated learning strategies and learning satisfaction. The mediation effects were generally significant (Table 8). The Sobel test was used for mediation analysis, where a Z-value greater than the absolute value of 1.65 indicated a significant mediation effect [83,86]. The indirect effect confidence intervals calculated using the product of the distribution method—when not including 0 at a 95% confidence level—indicate the presence of mediation effects. As the analytical results show, except for goal orientation through motivational strategies, the rest of the mediation effects are significantly influential on learning outcomes.

**Table 8.** Mediation effects of learning satisfaction (N = 380).

| Mediator Variable | | Path | | | Sobel Test's z-Value | | Product of Distribution | | | |
|---|---|---|---|---|---|---|---|---|---|---|
| | | | | | | | Mediation Effect | | LL 95% CI | UL 95% CI |
| H7 | Regulation of Cognition | LGO | →ROC→ | LS | 3.399 | *** | $\mu = 0.110$ *** | $(\sigma = 0.033)$ | 0.052 | 0.180 |
| | | PGO | →ROC→ | LS | 2.396 | * | $\mu = 0.046$ ** | $(\sigma = 0.020)$ | 0.013 | 0.090 |
| | | AGO | →ROC→ | LS | −2.534 | * | $\mu = -0.050$ ** | $(\sigma = 0.020)$ | −0.093 | −0.016 |
| | | LGO | →ROC→ | LO | 2.097 | * | $\mu = 0.110$ *** | $(\sigma = 0.031)$ | 0.054 | 0.176 |
| | | PGO | →ROC→ | LO | 1.782 | + | $\mu = 0.022$ + | $(\sigma = 0.013)$ | 0.002 | 0.051 |
| | | AGO | →ROC→ | LO | −1.836 | + | $\mu = -0.024$ + | $(\sigma = 0.013)$ | −0.053 | −0.002 |
| | Regulation of Motivation | LGO | →ROM→ | LS | 2.114 | * | $\mu = 0.052$ * | $(\sigma = 0.025)$ | 0.009 | 0.108 |
| | | PGO | →ROM→ | LS | 2.118 | * | $\mu = 0.044$ * | $(\sigma = 0.021)$ | 0.008 | 0.090 |
| | | AGO | →ROM→ | LS | −1.991 | * | $\mu = -0.037$ + | $(\sigma = 0.019)$ | −0.079 | −0.006 |
| | | LGO | →ROM→ | LO | 0.069 | | $\mu = 0.001$ | $(\sigma = 0.021)$ | −0.040 | 0.043 |
| | | PGO | →ROM→ | LO | 0.069 | | $\mu = 0.001$ | $(\sigma = 0.017)$ | −0.033 | 0.036 |
| | | AGO | →ROM→ | LO | −0.069 | | $\mu = -0.001$ | $(\sigma = 0.015)$ | −0.031 | 0.029 |
| | Regulation of Behavior | LGO | →ROB→ | LS | 2.902 | ** | $\mu = 0.110$ ** | $(\sigma = 0.038)$ | 0.042 | 0.192 |
| | | PGO | →ROB→ | LS | 2.665 | ** | $\mu = 0.067$ ** | $(\sigma = 0.026)$ | 0.023 | 0.123 |
| | | AGO | →ROB→ | LS | −2.094 | * | $\mu = -0.044$ * | $(\sigma = 0.022)$ | −0.093 | −0.009 |
| | | LGO | →ROB→ | LO | 2.667 | ** | $\mu = 0.096$ ** | $(\sigma = 0.036)$ | 0.031 | 0.173 |
| | | PGO | →ROB→ | LO | 2.479 | * | $\mu = 0.058$ ** | $(\sigma = 0.024)$ | 0.017 | 0.110 |
| | | AGO | →ROB→ | LO | −2.000 | * | $\mu = -0.038$ + | $(\sigma = 0.020)$ | −0.083 | −0.007 |
| H8 | Learning Satisfaction | ROC | →LS→ | LO | 3.802 | *** | $\mu = 0.137$ *** | $(\sigma = 0.036)$ | 0.069 | 0.211 |
| | | ROM | →LS→ | LO | 2.369 | ** | $\mu = 0.087$ ** | $(\sigma = 0.037)$ | 0.017 | 0.161 |
| | | ROB | →LS→ | LO | 3.128 | ** | $\mu = 0.137$ ** | $(\sigma = 0.044)$ | 0.054 | 0.227 |

Note 1: LGO = learning goal orientation; PGO = proving goal orientation; AGO = avoiding goal orientation; ROC = regulation of cognition; ROM = regulation of motivation; ROB = regulation of behavior; LS = learning satisfaction; LO = learning outcomes. Note 2: +, *p*-value < 0.1; *, *p*-value < 0.05; **, *p*-value < 0.01; ***, *p*-value < 0.001. Note 3: Number of bootstrap samples = 5000.

Integrating the above analysis, the goal orientations of the participants in this study mainly indirectly affect their learning satisfaction and learning outcomes through medi-

ating variables (self-regulated learning and learning satisfaction) (see Table 8). Therefore, H7: employees' goal orientations indirectly influence their learning satisfaction and learning outcomes through the mediating effects of self-regulated learning strategies, is mostly supported. H8: employees' self-regulated learning strategies indirectly influence their learning outcomes through the mediating effects of learning satisfaction, which is supported.

## 5. Discussion and Recommendations

### 5.1. Discussion

Based on the theory of goal orientation in educational psychology, this study found that, in an e-learning environment, the goal orientations held by corporate employees are related to their learning satisfaction through self-regulated learning. These orientations, in turn, indirectly affect their learning outcomes, with self-regulated learning and learning satisfaction as mediators. The empirical analysis and verification of our hypotheses can be explained as follows:

1.  The influence of goal orientation on self-regulated learning: The results of hypotheses H1abc, H2abc, and H3abc indicate that employees' goal orientations positively impact their self-regulated learning in an e-learning environment. Except for the avoidance goal orientation, which had a significant negative effect, the learning and proving goal orientations showed significant positive effects. This suggests that e-learning in a corporate environment, akin to traditional learning methods, is largely driven by motivation for achievement. Employees with a learning goal orientation typically seek self-improvement and skill enhancement, and thus opt for more in-depth learning strategies [8,38]. Employees with a proving goal orientation focus on comparative outcomes and aspire to demonstrate their competencies, hence they might adopt approaches akin to learning goal orientations but tend to utilize more superficial learning strategies for processing information to surpass others [8,38]. By contrast, those with an avoidance orientation, fearing failure, selectively abandon learning or employ fewer strategies [41,43,49]. These findings confirm that, in an e-learning setting, an individual's learning motivation and type of goal orientation are interconnected with their chosen self-regulated learning strategies, subsequently affecting their educational journey. The results of this study are consistent with and akin to past research [10,23,43,49,87].

2.  The effect of self-regulated learning on learning satisfaction: The analysis of hypothesis H4abc reveal that self-regulated learning positively influences learning satisfaction in an e-learning context, with all strategies significantly impacting satisfaction. This aligns with findings from previous research [50,53,61,62]. This study found that learners who effectively employ self-regulated learning strategies acquire enhanced knowledge and skills, leading to a sense of satisfaction [22]. These results not only align with related research [50,53,61] but also support the theoretical propositions of Artino [67]. These findings highlight the need for learners to adopt self-regulated learning strategies based on real-time situations in e-learning environments, promoting their satisfaction and increased engagement. The findings support the viewpoints of Dweck [43] and Elliot [87].

3.  The effects of self-regulated learning on learning outcomes: According to the analysis of hypothesis H5abc, self-regulated learning positively affects the learning outcomes in an e-learning environment. While motivational regulation did not have a significant positive impact, cognitive and behavioral regulations had significant positive effects. Several scholars have identified self-regulated learning as a critical predictor of learning outcomes [44,63]. This study's findings echo those of the past [50,61], asserting the positive relationship between self-regulated learning and learning outcomes. Unfortunately, the motivational aspect did not show a significant effect, possibly due to its reliance on intrinsic and extrinsic motivation strategies that the organization may not have effectively communicated to employees [45,46]. The results corroborate

past research assertions [43,49,51,52,66,87] and encourage further studies to validate the unconfirmed hypotheses.

4. The effects of learning satisfaction on learning outcomes: The analysis of hypothesis H6 reveals that learning satisfaction positively impacts learning outcomes in an e-learning environment. These results are consistent with the arguments of Kuo et al., [53] and Paechter, Maier, and Macher [88]. As proposed by Kim and Park [30], a strong correlation exists between learning satisfaction and outcomes. Higher satisfaction is associated with better outcomes and vice versa. Overall, learning satisfaction not only explains the motivation behind employee's participation and the results of their learning activities but also serves as a crucial indicator for gauging whether learners' outcomes and satisfaction needs are met [31,67]. Therefore, the empirical evidence from this study suggests that learning satisfaction significantly influences learning outcomes, supporting the propositions of Dweck [43] and Elliot [87].

5. The mediating effects of self-regulated learning and learning satisfaction: The analysis of hypotheses H7 and H8 reveals that, in an e-learning environment, goal orientations indirectly influence employees' learning outcomes by mediating self-regulated learning and learning satisfaction. Past research has identified the correlations between goal orientations, self-regulated learning, and learning outcomes, including satisfaction [22,52,64,67–70,80]. This study methodically deduced and established the interrelationships between goal orientations, self-regulated learning, learning satisfaction, and outcomes. The empirical findings indicate that the proposed framework is validated, showing that goal orientation primarily affects final learning outcomes through the mediation of self-regulated learning and learning satisfaction.

*5.2. Conclusions and Suggestions*

The feasibility, effectiveness, and sustainability of e-learning in enhancing employee self-learning, growth, and skill development have been topics of great interest to corporate department leaders, HR directors, top management, and scholars. This study confirms that e-learning is a practical and effective education technology and training method for employees and that it can serve as a strategic learning model in organizations. Our research highlights that learning and proving goal orientations positively influence employee engagement in self-regulated learning, whereas an avoidance goal orientation tends to have a detrimental effect. Furthermore, self-regulated learning substantially boosts learning satisfaction, positively impacting learning outcomes. Thus, an individual's learning satisfaction directly correlates with their learning performance. Based on these findings, this study offers the following recommendations for managers, HR departments, and senior executives:

1. Guiding employee goal orientations: It has been demonstrated that learners who exhibit learning and proving goal orientations show a positive relationship with self-regulated learning. Hence, in terms of educational and training initiatives, it is vital to understand and align them with the goal orientations of employees. Encouraging positive learning motivation and orientation can motivate employees toward learning-focused goals, thereby enhancing their learning capabilities.

2. Assisting employees in developing effective self-regulated learning strategies: Learners often recognize various aspects of their self-regulated learning during training, including cognitive, motivational, and behavioral elements. Providing timely assistance and facilitating discussions about learning processes can help employees acknowledge the effectiveness of their current learning strategy and identify areas for improvement. Guiding employees in adapting and executing suitable self-regulated learning strategies throughout their journey can lead to optimal educational outcomes.

3. Creating a sustainability and learning-valuing environment: In today's rapidly changing and competitive economic landscape, organizations depend on continual employee learning to maintain their competitive edge and sustainable development. Systematically creating a conducive learning environment not only fosters motivation

aligned with learning goals but also enables the adoption of effective learning methods, leading to increased satisfaction and the achievement of learning objectives. A high-quality learning environment is integral to positive employee development.

4. Optimizing e-learning platforms for comprehensive learning monitoring and feedback: Encouraging employees to utilize e-learning platforms effectively, with features such as progress tracking, skill assessment, goal achievement rates, and course discussions/feedback, can greatly enhance their training experience. The roles of self-regulated learning and learning satisfaction, as mediators, suggest that the proficient use of e-learning platforms enhances job skills and facilitates flexible learning methods, resulting in more satisfactory learning outcomes.

5. Wholehearted support for education technology from senior management: Senior management's commitment and active involvement in promoting e-learning are crucial. Establishing motivational mechanisms can encourage employee engagement. Continuous support from top-level management can enhance employees' positive learning attitudes and willingness to participate in training, which is critical to successfully integrating e-learning into organizational operations. Decision-makers should recognize the skills and performance enhancements provided by e-learning and value the use of tools and feedback processes, aligning them with organizational goals and strategies to increase business value.

Despite our rigorous efforts, this study has limitations, which call for further exploration and discussion. The following points highlight these limitations and suggest directions for future research:

1. Exploring additional influencing factors: In our quest for a concise framework, we examined the influence of personal goal orientations on self-regulated learning, learning satisfaction, and the outcomes of employees. However, many factors could influence the experiences and outcomes of employees following e-learning.

2. Expanding the industry sample: This study was confined to employees in the manufacturing and trading sectors who engaged in e-learning. It did not include employees from the service sector (e.g., the life insurance and banking industries), which may limit the generalizability of our findings.

3. Assessment of differences in learning outcomes: The learning outcomes in this study were measured through self-evaluation by employees post e-learning. This approach might differ from traditional classroom-based educational training assessments, and this study did not explore these potential differences.

4. Long-term impact on job performance: While past research underscores post-learning job performance as a critical aspect of learning outcomes, this cannot be effectively measured in the short term and requires medium- to long-term evaluation. This study focused on individuals' perceived outcomes, and did encompass a longer-term assessment.

5. Cross-sectional data limitations: The collected data were cross-sectional, which may be needed to fully capture the dynamic nature of learning processes and outcomes over time.

6. Sampling restrictions: The questionnaire utilized a convenience sampling method, which, although easy to implement and convenient for accessing samples, might introduce sampling bias.

It is recommended that future researchers make improvements in the following directions:

1. Incorporating broader factors: Future research could explore the influence of additional elements like personal beliefs and the organizational environment on goal orientation, leadership styles, and their impact on the goal orientations of employees. Future studies could also investigate the relationship between personal goal orientations and goal setting, reactions to performance feedback, and feedback-seeking behavior.

2. Industry diversification: Expanding the sample to include service industries (e.g., life insurance, banking), as well as larger manufacturing and distribution sectors, could enhance the robustness and applicability of these findings.

3. Comparative studies on learning modalities: Studies comparing outcomes between traditional education and e-learning or evaluating the effectiveness of blended learning approaches could be conducted.

4. Longitudinal studies on learning outcomes: Future research could examine the medium- to long-term impact of learning outcomes for individuals with different goal orientations, assessing the sustainability of these outcomes over time.

5. Longitudinal approach and diverse assessment methods: Adopting a longitudinal study design to gather data at different intervals could offer insights into the evolution of learning behaviors and outcomes. Alongside self-evaluations, incorporating assessments from e-learning systems or managerial evaluations could provide a more objective view of employees' learning achievements.

6. Improve sampling methods: To ensure the sample's representativeness, future studies could increase and expand the diversity of the sample. Collecting and analyzing the demographic data of the sample and comparing it with the overall demographic data of the organization will help identify the sample's representativeness.

**Author Contributions:** Conceptualization, C.-Y.T.; methodology, C.-Y.T.; software, C.-Y.T.; validation, C.-Y.T.; formal analysis, C.-Y.T.; investigation, C.-Y.T.; resources, C.-Y.T.; data curation, C.-Y.T.; writing—original draft preparation, C.-Y.T.; writing—review and editing, C.-Y.T.; visualization, C.-Y.T.; supervision, D.-C.L.; project administration, C.-Y.T. and D.-C.L. All authors have read and agreed to the published version of the manuscript.

**Funding:** This research received no external funding.

**Institutional Review Board Statement:** Not applicable.

**Informed Consent Statement:** Informed consent was obtained from all subjects involved in the study.

**Data Availability Statement:** The data presented in this study are available on request from the corresponding author.

**Acknowledgments:** The authors would like to thank the Editor and the anonymous reviewers for their thoughtful and constructive comments that have greatly improved this manuscript.

**Conflicts of Interest:** The authors declare no conflicts of interest.

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
