# Peer review of "Enterprise Implementation of Educational Technology: Exploring Employee Learning Behavior in E-Learning Environments"

_sustainability, doi:10.3390/su16041679_

Round 1

Reviewer 1 Report

Comments and Suggestions for Authors

Grounded in a corporate e-learning environment facilitated by educational technology, this study delves into and formulates hypotheses on the impact of employee achievement motivations, encompassing learning, proving, and avoiding goal orientations, on self-regulated learning. This includes cognitive, motivational, and behavioral adjustments. Additionally, the research investigates how employees' self-regulated learning and satisfaction with the learning process influence the assessment of learning effectiveness. The study, involving 380 employees engaged in corporate educational technology learning, unveils that learners' orientations indirectly influence ultimate learning outcomes. This occurs through the mediating roles of self-regulated learning and satisfaction. In summary, the paper is well-motivated, and further improvements can be made in the following areas:

(1) Section 1 could explicitly highlight the study's motives for exploring employee learning behavior and its correlations in an e-learning environment, itemizing each motivation.

(2) For clarity, Section 3.3 could include a simple illustrative form of the questionnaire design.

(3) In Section 4.3, results could be succinctly summarized in a table for better comprehension.

(4) The overall study logic can be bolstered by comparing it with similar studies in the case analysis, incorporating additional quantitative analysis where relevant.

(5) Enhancements to the presentation style, such as using italic style for variables, could improve overall readability.

Comments on the Quality of English Language

Minor editing of English language required.

Author Response

Response to Reviewer 1’s Comments

Grounded in a corporate e-learning environment facilitated by educational technology, this study delves into and formulates hypotheses on the impact of employee achievement motivations, encompassing learning, proving, and avoiding goal orientations, on self-regulated learning. This includes cognitive, motivational, and behavioral adjustments. Additionally, the research investigates how employees' self-regulated learning and satisfaction with the learning process influence the assessment of learning effectiveness. The study, involving 380 employees engaged in corporate educational technology learning, unveils that learners' orientations indirectly influence ultimate learning outcomes. This occurs through the mediating roles of self-regulated learning and satisfaction. In summary, the paper is well-motivated, and further improvements can be made in the following areas:

Response:

First of all, thank you very much for your encouraging and inspiring feedback on my work and for your constructive and helpful comments that have greatly improved the paper. I studied all of your comments carefully and tried to incorporate them into this article's current version. Please find more detailed descriptions of how I did that below. We have modified our research to accommodate your concerns (significant changes in the article are highlighted in red font with a yellow background for your quick perusal). In addition, we provided a point-by-point response to each of your comments. For ease of distinction, your original comments are listed below in regular font, and our responses to those comments are shown in italics font.

(1) Section 1 could explicitly highlight the study's motives for exploring employee learning behavior and its correlations in an e-learning environment, itemizing each motivation.

Response:

The content of 1. Introduction has been rewritten based on the suggestions.

For example, page 1 to page 2 of the PDF file content, lines 34 to 93.

(2) For clarity, Section 3.3 could include a simple illustrative form of the questionnaire design.

Response:

The content of 3.4. The Questionnaire Design has been adjusted based on the suggestions.

For example, on page 10, lines 421 to 425.

Table 3. Construct and References.

Construct

Research Variable

Items

References

Goal orientation

learning goal orientation

7

VandeWalle [43]

Elliot and Church [42]

Pintrich [22]

proving goal orientation

6

avoiding goal orientation

5

Self-regulated learning

cognitive regulation

6

Pintrich [22]

Gordon et al., [56]

Bouffard et al., [57]

motivational regulation

6

behavioral regulation

6

Learning satisfaction

6

Kuo et al., [65]

Learning outcomes

5

Alavi et al., [75]

Pike et al., [72]

Demographic variables

(3) In Section 4.3, results could be succinctly summarized in a table for better comprehension.

Response:

The content of 4.3. Verification of Hypotheses has been rewritten and adjusted based on the suggestions.

For example, on page 14, lines 522 to 534.

(4) The overall study logic can be bolstered by comparing it with similar studies in the case analysis, incorporating additional quantitative analysis where relevant. 

Response:

Thanks to the reviewer for the reminder!

We have rewritten 1. Introduction, 2. Literature Review, 3. Hypothesis... and other content based on the suggestions and opinions of other reviewers. We hope to respond to your suggestions.

(5) Enhancements to the presentation style, such as using italic style for variables, could improve overall readability.

Response:

Thanks for the advice!

Changing to italics can improve readability. Regarding the recommendation and checking the relevant regulations based on evidence, The current journal regulations do not recommend using this method of presentation.

Reviewer 2 Report

Comments and Suggestions for Authors

11.      I would like to recommend the authors to make the title more succinct. For example, “Enterprise Implementation of Educational Technology: Exploring Employee Learning Behavior in E-learning Environments” would be more concise and clearer to deliver the major topic/theme of the paper.

22.      While the abstract provides a general overview of the research, there are some areas need to be developed:

11)     Lack of Specifics: The abstract is somewhat vague about the methodology used in the study. It mentions the study was conducted with 380 employees but does not delve into details regarding the research design, data collection methods, or analysis techniques.

22)     Absence of Key Findings: The abstract mentions the study's exploration and hypothesis but does not highlight specific findings or results. Including a sentence or two on the key outcomes of the research would provide a clearer picture of the study's contributions.

33.      In line 48, could you specify what “HR” experts stand for? I would be great if you could give an original version of the acronym.

44.      The paragraph (line 74-78) is just same as the previous paragraph (line 69-73), which needs to be removed.

55.      Your research questions: this research explores how goal orientation in a e-learning environment affects self-regulated learning by employees and how these strategies, in turn, impact learning satisfaction and outcomes.

Based on the provided research questions, the following subtitles for the literature review could be considered, and I suggest reorganizing the literature review into three subsections:

11)     Goal Orientation in E-Learning Environments

Review literature related to goal orientation theories in educational psychology.

Examine studies that focus on the impact of goal orientation on learners in e-learning environments.

Discuss relevant research on goal orientation in workplace e-learning settings.

22)     Self-Regulated Learning in Employee Education

Investigate the concept of self-regulated learning, its theoretical foundations, and its relevance in corporate education.

Review studies exploring how employees engage in self-regulated learning, particularly within e-learning environments.

Examine factors influencing self-regulated learning in professional development settings.

33)     Linking Learning Satisfaction and Outcomes to Goal Orientation and Self-Regulated Learning

Explore literature on the relationship between goal orientation, self-regulated learning, and learning satisfaction.

Review studies that investigate the impact of self-regulated learning strategies on learning outcomes.

Examine the interplay between goal orientation, self-regulated learning, and overall learning effectiveness in e-learning contexts.

Due to the presence of nine subsections within the literature review, there is a concern that it might distract readers. Therefore, I highly recommend reorganizing and refining the literature review to enhance its coherence. (As noted above) While some of the suggested items under the three subsection have been adequately addressed, there are gaps in coverage that need attention. It is crucial to carefully review the items (suggested above) and incorporate any missing elements.

Furthermore, I acknowledge the inclusion of hypotheses under each subsection to provide clarity on the research hypothesis. However, as a reader, I find it challenging to navigate through the literature review. For improved clarity, it is suggested to relocate the hypotheses to the methodology section. This adjustment will streamline the literature review and contribute to a more cohesive and reader-friendly presentation of your research.

66.      The content within the conclusions (lines 556-624) appears to resemble more of the discussions of your study rather than the conclusions. Therefore, I propose renaming the section currently labeled as "Conclusions" (lines 556-624) to "Discussion." This adjustment accurately reflects the content and ensures a clearer distinction between the discussions and the conclusive elements of the study. Additionally, to enhance the structure and coherence of the document, it is recommended to merge sections 5.2 Management Implications and 5.3 Limitations and Future Research Directions. This consolidation can effectively serve as the conclusions of the study, providing a comprehensive and integrated summary of the managerial implications and areas for future research. This adjustment will contribute to a more streamlined and logically organized conclusion section in your study.

Comments on the Quality of English Language

Moderate editing of English language required

Author Response

Response to Reviewer 2’s Comments

First of all, thank you very much for your encouraging and inspiring feedback on my work and for your constructive and helpful comments that have greatly improved the paper. I studied all of your comments carefully and tried to incorporate them into this article's current version. Please find more detailed descriptions of how I did that below. We have modified our research to accommodate your concerns (significant changes in the article are highlighted in red font with a yellow background for your quick perusal). In addition, we provided a point-by-point response to each of your comments. For ease of distinction, your original comments are listed below in regular font, and our responses to those comments are shown in italics font.

  1. I would like to recommend the authors to make the title more succinct. For example, “Enterprise Implementation of Educational Technology: Exploring Employee Learning Behavior in E-learning Environments” would be more concise and clearer to deliver the major topic/theme of the paper.

Response:

 Thanks to the reviewer's suggestion, the title has been corrected.

  1. While the abstract provides a general overview of the research, there are some areas need to be developed:

1)     Lack of Specifics: The abstract is somewhat vague about the methodology used in the study. It mentions the study was conducted with 380 employees but does not delve into details regarding the research design, data collection methods, or analysis techniques.

2)     Absence of Key Findings: The abstract mentions the study's exploration and hypothesis but does not highlight specific findings or results. Including a sentence or two on the key outcomes of the research would provide a clearer picture of the study's contributions.

Response:

Thanks for the suggestion. The summary has been rewritten.

Such as the following content:

" The empirical survey targeted 380  employees from  26  companies participating in corporate educational technology learning  (e-earning),  with the research hypotheses tested through PLS structural equation modeling. The analysis indicates that employees' learning and proving goal orientations indirectly positively affect learning outcomes by mediating self-regulated learning and learning satisfaction. Conversely, employees' avoidance goal orientation indirectly negatively impacts learning outcomes by mediating self-regulated learning and learning satisfaction. Finally, the researchers offer recommendations for management implications and future research directions. "

  1. In line 48, could you specify what “HR” experts stand for? I would be great if you could give an original version of the acronym.

Response:

Thank you for reminding me that the full name has been added before the abbreviation. Such as line 52 on page 2. As follows:

"...training and human resources (HR) experts, and scholars. "

  1. The paragraph (line 74-78) is just same as the previous paragraph (line 69-73), which needs to be removed.

Response:

Thanks for reminding me that the paragraph about lines 69-73 is repeated in lines 74-78. I have checked the entire text during rewriting to ensure this problem will not occur again.

  1. Your research questions: this research explores how goal orientation in a e-learning environment affects self-regulated learning by employees and how these strategies, in turn, impact learning satisfaction and outcomes.

Based on the provided research questions, the following subtitles for the literature review could be considered, and I suggest reorganizing the literature review into three subsections:

1)     Goal Orientation in E-Learning Environments

Review literature related to goal orientation theories in educational psychology.

Examine studies that focus on the impact of goal orientation on learners in e-learning environments.

Discuss relevant research on goal orientation in workplace e-learning settings.

2)     Self-Regulated Learning in Employee Education

Investigate the concept of self-regulated learning, its theoretical foundations, and its relevance in corporate education.

Review studies exploring how employees engage in self-regulated learning, particularly within e-learning environments.

Examine factors influencing self-regulated learning in professional development settings.

3)     Linking Learning Satisfaction and Outcomes to Goal Orientation and Self-Regulated Learning

Explore literature on the relationship between goal orientation, self-regulated learning, and learning satisfaction.

Review studies that investigate the impact of self-regulated learning strategies on learning outcomes.

Examine the interplay between goal orientation, self-regulated learning, and overall learning effectiveness in e-learning contexts.

Due to the presence of nine subsections within the literature review, there is a concern that it might distract readers. Therefore, I highly recommend reorganizing and refining the literature review to enhance its coherence. (As noted above) While some of the suggested items under the three subsection have been adequately addressed, there are gaps in coverage that need attention. It is crucial to carefully review the items (suggested above) and incorporate any missing elements.

Furthermore, I acknowledge the inclusion of hypotheses under each subsection to provide clarity on the research hypothesis. However, as a reader, I find it challenging to navigate through the literature review. For improved clarity, it is suggested to relocate the hypotheses to the methodology section. This adjustment will streamline the literature review and contribute to a more cohesive and reader-friendly presentation of your research.

 Response:

Once again, we would like to thank the reviewers for their hard work and for providing suggestions for improving this study. We will address your comments item by item.

1) The suggestions have been adopted to reorganize and rewrite the content 2. Literature Review,

Details are shown on pages 2 to 7, lines 94 to 308, because the entire chapter has been rewritten.

Please refer to the paper.

2) The suggestions for hypotheses have been rewritten and moved to section 3.1. Hypotheses of 3. Hypotheses and Research Methodology.

Moreover, appropriately modify the architecture diagram and subsequent related hypothesis verification content.

For details, please refer to the paper on pages 7 to 8, lines 309 to 367, as the entire chapter has been rewritten.

  1. The content within the conclusions (lines 556-624) appears to resemble more of the discussions of your study rather than the conclusions. Therefore, I propose renaming the section currently labeled as "Conclusions" (lines 556-624) to "Discussion." This adjustment accurately reflects the content and ensures a clearer distinction between the discussions and the conclusive elements of the study. Additionally, to enhance the structure and coherence of the document, it is recommended to merge sections 5.2 Management Implications and 5.3 Limitations and Future Research Directions. This consolidation can effectively serve as the conclusions of the study, providing a comprehensive and integrated summary of the managerial implications and areas for future research. This adjustment will contribute to a more streamlined and logically organized conclusion section in your study.

 Response:

Thanks for the comments! The section dealing with the Conclusions has been renamed Discussion. For the suggestion it was merged with 5.2 Management Implications and 5.3 Limitations and Future Research Directions.

For details, please refer to pages 15 to 19, lines 567 to 737, and rewrite the key points accordingly.

Please refer to the paper.

Reviewer 3 Report

Comments and Suggestions for Authors

The overall quality of the paper is commendable, particularly in its adept application of learning theories to the context of workplace training. However, there is a need for a more comprehensive elucidation of the methods employed, with a specific emphasis on providing a detailed exposition of the questionnaire measure utilized. A more intricate examination of the questionnaire, its design, and the rationale behind its selection would substantially enhance the methodological clarity.

The inclusion of demographic information in the presentation raises questions, as these aspects are not thoroughly explored within the body of the paper. A more robust justification for the collection and presentation of demographic data is warranted to align with the overarching goals and focus of the study. Moreover, delving into the participants' perspectives on the mandatory or optional nature of the training, as well as their perception of its relevance, would offer valuable insights into the nuanced aspects of workplace learning.

Consideration of the mediating role of learning approaches could elevate the analytical depth of the study. Exploring how various learning approaches may act as mediators in the relationship between the applied learning theories and the effectiveness of workplace training would contribute a nuanced layer to the findings.

The discussion section provides an opportunity for further expansion, particularly in addressing limitations, delineating implications, and suggesting future research directions. A more thorough examination of the study's constraints will contribute to a balanced interpretation of the findings. Furthermore, outlining the practical implications of the research outcomes for workplace training initiatives and organizational practices will enhance the paper's real-world applicability. Lastly, offering concrete suggestions for future research avenues within the domain of learning theories and workplace training will underscore the scholarly contribution and potential avenues for further exploration in the field.

Comments on the Quality of English Language

in couple of places this would benefit from some slight tweaking to improve its clarity.

Author Response

Response to Reviewer 3’s Comments

The overall quality of the paper is commendable, particularly in its adept application of learning theories to the context of workplace training. However, there is a need for a more comprehensive elucidation of the methods employed, with a specific emphasis on providing a detailed exposition of the questionnaire measure utilized. A more intricate examination of the questionnaire, its design, and the rationale behind its selection would substantially enhance the methodological clarity.

The inclusion of demographic information in the presentation raises questions, as these aspects are not thoroughly explored within the body of the paper. A more robust justification for the collection and presentation of demographic data is warranted to align with the overarching goals and focus of the study. Moreover, delving into the participants' perspectives on the mandatory or optional nature of the training, as well as their perception of its relevance, would offer valuable insights into the nuanced aspects of workplace learning.

Consideration of the mediating role of learning approaches could elevate the analytical depth of the study. Exploring how various learning approaches may act as mediators in the relationship between the applied learning theories and the effectiveness of workplace training would contribute a nuanced layer to the findings.

The discussion section provides an opportunity for further expansion, particularly in addressing limitations, delineating implications, and suggesting future research directions. A more thorough examination of the study's constraints will contribute to a balanced interpretation of the findings. Furthermore, outlining the practical implications of the research outcomes for workplace training initiatives and organizational practices will enhance the paper's real-world applicability. Lastly, offering concrete suggestions for future research avenues within the domain of learning theories and workplace training will underscore the scholarly contribution and potential avenues for further exploration in the field.

 Response:

First of all, thank you very much for your encouraging and inspiring feedback on my work and for your constructive and helpful comments that have greatly improved the paper. I studied all of your comments carefully and tried to incorporate them into this article's current version. Please find more detailed descriptions of how I did that below. We have modified our research to accommodate your concerns (significant changes in the article are highlighted in red font with a yellow background for your quick perusal). In addition, we provided a point-by-point response to each of your comments. For ease of distinction, your original comments are listed below in regular font, and our responses to those comments are shown in italics font.

Since the relevant suggestions are similar to those of other reviewers, we have integrated related issues and rewritten the entire paper.

Briefly, regarding the significant revisions made as per the reviewers' suggestions, highlighted with red font and a yellow background:

The title has been revised.

The abstract has been appropriately modified.

The introduction has been rewritten.

The literature review has been revised.

The Hypotheses and Research Methodology section has been appropriately rewritten.

Specific Results section charts have been corrected.

Titles and descriptions in the Results section have been amended.

References have been reorganized.

Reviewer 4 Report

Comments and Suggestions for Authors

The paper is outstanding and can be published. Although, I want to make some minor recommendations that could help authors to improve it. Specifically, the authors have successfully tried to address which is the impact of employee achievement motivations on self-regulated learning. Furthermore, the manuscript explores how workers’ self-regulated learning and learning satisfaction with the learning process influence learning effectiveness assessment.

The authors’ approach is innovative. The whole research project is relevant and confirms that an e-learning is a practical and effective educational technology and training method for the workforce and may offer a strategic learning model within enterprises.

More specifically, this paper is scientifically sound, and the research design is valid. The conclusions are strongly linked with the evidence. All hypotheses have been addresses and discussed.  The ethics statement is adequate. The references are relevant. However, several sources are quite old. Is this necessary or authors could use more recent references?

Novelty: The manuscript is novel, and advances adequately the knowledge. The manuscript offers to the knowledge, especially by offering specific practical recommendation concerning management implications.   

The paper fits the scope of the journal. The significance of the paper is high enough. Tables and figures have been used properly and provide a better comprehension of the collected evidence and serve the assessment of research questions/hypotheses. The quality of the paper is good enough, as it has been written appropriately.

This manuscript will interest the readers. The overall merit is positive, the publication will bring about benefits in knowledge. The manuscript may be accepted with minor revisions. The paragraph in lines 69-73 has been repeated in lines 74-78. An amendment is needed.

In many lines the authors have mentioned the publishing year before the reference i.e. in line 69 “ Kellers (1983) [29]”. Is this appropriate according to the author guidelines of the journal?

Lastly, the authors should add some more information, such as a) the total population of workers of the survey; b) explain whether the sample reflects properly the total workforce of the 26 organizations; c) when the survey has been contacted?     

Author Response

Response to Reviewer 4’s Comments

First of all, thank you very much for your encouraging and inspiring feedback on my work and for your constructive and helpful comments that have greatly improved the paper. I studied all of your comments carefully and tried to incorporate them into this article's current version. Please find more detailed descriptions of how I did that below. We have modified our research to accommodate your concerns (significant changes in the article are highlighted in red font with a yellow background for your quick perusal). In addition, we provided a point-by-point response to each of your comments. For ease of distinction, your original comments are listed below in regular font, and our responses to those comments are shown in italics font.

The paper is outstanding and can be published. Although, I want to make some minor recommendations that could help authors to improve it. Specifically, the authors have successfully tried to address which is the impact of employee achievement motivations on self-regulated learning. Furthermore, the manuscript explores how workers’ self-regulated learning and learning satisfaction with the learning process influence learning effectiveness assessment.

 The authors’ approach is innovative. The whole research project is relevant and confirms that an e-learning is a practical and effective educational technology and training method for the workforce and may offer a strategic learning model within enterprises.

 More specifically, this paper is scientifically sound, and the research design is valid. The conclusions are strongly linked with the evidence. All hypotheses have been addresses and discussed.  The ethics statement is adequate. The references are relevant. However, several sources are quite old. Is this necessary or authors could use more recent references?

 Novelty: The manuscript is novel, and advances adequately the knowledge. The manuscript offers to the knowledge, especially by offering specific practical recommendation concerning management implications.   

 The paper fits the scope of the journal. The significance of the paper is high enough. Tables and figures have been used properly and provide a better comprehension of the collected evidence and serve the assessment of research questions/hypotheses. The quality of the paper is good enough, as it has been written appropriately.

Response:

Thanks for your valuable advice! Regarding the use of some earlier references, it should be noted that these materials will be cited as their primary sources in this field. Thanks for your careful review carefully and suggestions. In the process of this rewriting, the citations of the literature have been readjusted.

This manuscript will interest the readers. The overall merit is positive, the publication will bring about benefits in knowledge. The manuscript may be accepted with minor revisions. The paragraph in lines 69-73 has been repeated in lines 74-78. An amendment is needed.

Response:

Thanks for reminding me that the paragraph about lines 69-73 is repeated in lines 74-78. I have checked the entire text during rewriting to ensure this problem will not occur again.

In many lines the authors have mentioned the publishing year before the reference i.e. in line 69 “ Kellers (1983) [29]”. Is this appropriate according to the author guidelines of the journal?

Response:

Thanks for reminding me that the important question. We have checked the entire text during rewriting to ensure this problem will not occur again.

Lastly, the authors should add some more information, such as a) the total population of workers of the survey; b) explain whether the sample reflects properly the total workforce of the 26 organizations; c) when the survey has been contacted?     

Response:

Thanks for your insightful comment.

Regarding this section, the content related to 3.3 Research Participants and Data Collection has been rewritten. The detailed content is from line 341 to 389, on pages 7 to 8. The modifications are summarized below:

a) "Most respondents came from the manufacturing industry, accounting for 53.85%. Regarding department affiliation among the 380 employees, administrative, manufacturing, business, and shipping departments represented 10.00%, 18.68%, 14.47%, and 12.63%, respectively, with a total response rate across these departments of 55.78%. The sample structure of the companies and the departments of the employees are detailed in Table 1."

b) The revised content includes a statement on the limitations of the sampling method and remedial measures: "Given the convenience sampling method, there is a potential risk of sampling bias. However, the diversity in the sample source (departments, gender, age groups, work experience, etc.), as shown in Tables 1 and 2, suggests that the sample is diverse. Further statistical analyses (T-tests and ANOVA) showed no significant differences in perceptions of the study variables among participants from different sources. Lastly, early respondents (those from the first week) and late respondents (those from the fourth week) were compared to detect any differences in perceptions of the study variables, with the analysis showing no significant differences. Overall, the returned questionnaires are considered to have appropriate representativeness."

c) "The survey was initiated on July 1, 2023, and concluded on July 31, 2023, with data collected anonymously. A total of 800 questionnaires were distributed across 40 companies, averaging about 20 per company."